# Unveiling Mucopolysaccharidosis IIIC in Brazil: Diagnostic Journey and Clinical Features of Brazilian Patients Identified Through the MPS Brazil Network

**DOI:** 10.3390/diseases14010005

**Published:** 2025-12-26

**Authors:** Yorran Hardman Araújo Montenegro, Maria Fernanda Antero Alves, Simone Silva dos Santos-Lopes, Carolina Fischinger Moura de Souza, Fabiano de Oliveira Poswar, Ana Carolina Brusius-Facchin, Fernanda Bender-Pasetto, Kristiane Michelin-Tirelli, Fernanda Medeiros Sebastião, Franciele Barbosa Trapp, Erlane Marques Ribeiro, Paula Frassinetti Vasconcelos de Medeiros, Chong Ae Kim, Emilia Katiane Embiraçu, Mariluce Riegel-Giugliani, Guilherme Baldo, Roberto Giugliani

**Affiliations:** 1Instituto Nacional de Genética Médica e Populacional (INAGEMP), Hospital de Clínicas de Porto Alegre, Porto Alegre 90035-903, Brazil; 2MPS Brazil Network, Hospital de Clínicas de Porto Alegre, Porto Alegre 90035-903, Brazil; 3Laboratório de Ancestralidade e Biologia Molecular (LABMOL), Universidade Estadual da Paraíba, Campina Grande 58429-500, Brazil; 4Serviço de Genética Médica, Hospital de Clínicas de Porto Alegre, Porto Alegre 90035-903, Brazil; 5Casa dos Raros, Porto Alegre 90610-261, Brazil; 6Programa de Pós-Graduação em Genética e Biologia Molecular, Universidade Federal do Rio Grande do Sul, Porto Alegre 91501-970, Brazil; 7Hospital Infantil Albert Sabin, Fortaleza 60410-794, Brazil; 8Unidade Acadêmica de Medicina, Hospital Universitário Alcides Carneiro, Universidade Federal de Campina Grande, Campina Grande 58109-900, Brazil; 9Genetics Unit, Departamento f Pediatrics, Faculdade de Medicina, Universidade de São Paulo, São Paulo 05508-000, Brazil; 10Departamento de Medicina, Universidade Federal da Bahia, Salvador 40110-903, Brazil; 11Células, Tecidos e Genes, Hospital de Clínicas de Porto Alegre, Porto Alegre 90035-003, Brazil; 12DASA Genômica, São Paulo 06455-010, Brazil

**Keywords:** Sanfilippo C syndrome, mucopolysaccharidosis type IIIC, HGSNAT, heparan sulfate, Brazil

## Abstract

Background: Mucopolysaccharidosis type IIIC (MPS IIIC) is a rare lysosomal storage disorder caused by pathogenic variants in the HGSNAT gene. Data from large patient cohorts remain scarce, particularly in Latin America. Methods: We retrospectively analyzed clinical, biochemical, and genetic data from patients diagnosed with MPS IIIC through the MPS Brazil Network. Diagnosis was based on reduced activity of acetyl-CoA:α-glucosaminide N-acetyltransferase (HGSNAT), elevated urinary glycosaminoglycans (uGAGs), and/or molecular genetics tests. Results: A total of 101 patients were confirmed with MPS IIIC, representing one of the largest cohorts worldwide. Females accounted for 60% of cases. The mean age at symptom onset was 5.4 ± 3.9 years, while the mean age at diagnosis was 11.7 ± 6.9 years, reflecting a 6-year diagnostic delay. Most patients initially presented with developmental delay (82%) and facial dysmorphism (80%), whereas behavioral manifestations were less frequently identified (25%), suggesting a milder phenotype than previously reported. Genetic information was available for 28% of patients, showing recurrent alleles (c.372-2A>G, c.252dupT) and several novel mutations, which expand the mutational spectrum of the disease. Genotype–phenotype similarities with Portuguese, Italian, and Chinese cases suggest shared ancestry contributions. Regional differences included earlier diagnoses in the North of Brazil and high consanguinity rates in the Northeast region. Conclusions: This study describes the largest Brazilian cohort of MPS IIIC, documenting novel variants and regional heterogeneity. Findings highlight diagnostic delays, ancestry influences, and the urgent need for disease-modifying therapies.

## 1. Introduction

Mucopolysaccharidoses (MPS) are a group of lysosomal storage disorders primarily caused by deficiencies in enzymes responsible for the degradation of glycosaminoglycans (GAGs), formerly known as mucopolysaccharides [1]. The MPS are classified based on the presence of a specific enzyme deficiency underlying each disorder. In this work, we focus on Mucopolysaccharidosis type III C (MPS IIIC), a subtype of Mucopolysaccharidosis III (MPS III), or Sanfilippo Syndrome, characterized by impaired degradation of heparan sulphate [2].

Mutations in distinct genes lead to four subtypes of MPS III: MPS IIIA, caused by mutations in the *SGSH* gene, which encodes the enzyme N-sulfoglucosamine sulfohydrolase (EC 3.10.1.1; OMIM #252900); MPS IIIB, caused by mutations in the *NAGLU* gene, which encodes the enzyme N-alpha-acetylglucosaminidase (EC 3.2.1.50; OMIM #252920); MPS IIIC, caused by mutations in the *HGSNAT* gene, which encodes the enzyme heparan acetyl-CoA:alpha-glucosaminide N-acetyltransferase (EC 2.3.1.78; OMIM #252930); and MPS IIID, caused by mutations in the *GNS* gene, which encodes the enzyme N-acetylglucosamine-6-sulfatase (EC 3.1.6.14; OMIM #252940) [3].

All subtypes of MPS III present with a similar clinical course. During the first six years of life, the predominant manifestations include delayed neuropsychomotor development and behavioral disturbances, often leading to misdiagnoses such as Autism Spectrum Disorder [4]. Between six and ten years of age, patients typically experience marked cognitive decline accompanied by progressive behavioral worsening [5]. After the age of ten, behavioral problems tend to decrease as severe neurodegeneration becomes established, ultimately leading to premature death, usually in the third decade of life [6].

Therapeutic management of MPS IIIC remains a major challenge in clinical practice. Approaches such as hematopoietic stem cell transplantation using the native enzyme have shown limited or no success, and enzyme replacement therapy has likewise failed to achieve meaningful clinical benefits [7]. Despite these limitations, significant efforts have been directed toward developing effective alternatives for affected patients. Promising advances have been reported in the use of small molecules, including substrate reduction therapies [8,9], pharmacological chaperones to correct enzyme misfolding [10], anti-inflammatory strategies [11], inhibitors of protein aggregation to mitigate neurological damage (Monaco et al., 2020), and agents targeting oxidative stress [12]. Nevertheless, a major hope in the therapeutic landscape of MPS IIIC has emerged from gene therapy, which currently represents the most promising avenue for a disease-modifying treatment [7].

As information on MPS IIIC is scarce, this study provides a unique opportunity to advance the understanding of this pathology, particularly given the relatively large sample size identified in Brazil. Accordingly, our objective was to conduct a comprehensive survey of diagnostic data for MPS IIIC from 1983 to 2025, supported by the MPS Brazil Network. The MPS Brazil Network serves as a major reference center for the diagnosis of mucopolysaccharidoses in Brazil, offering biochemical analyses of enzymes and biomarkers by traditional and innovative methods such as tandem mass spectrometry [13]. Additional services include molecular analyses to identify pathogenic variants, as well as guidance in referring patients to specialized centers close to their home [14]. This paper reflects the extensive support our center has provided to physicians and patients throughout Brazil and Latin America for over four decades for the identification of MPS IIIC cases.

## 2. Methods

### 2.1. Retrospective Study

A retrospective study was conducted on patients diagnosed with MPS IIIC who were born between 1983 and 2025 and diagnosed by the MPS Brazil Network, operating at Hospital de Clínicas de Porto Alegre (HCPA) and at Casa dos Raros (Porto Alegre, Brazil, Rio Grande do Sul). To be included in the study, patients were required to demonstrate biochemical evidence of HGSNAT-deficient activity, increased urinary glycosaminoglycans (uGAGs) with predominance of heparan sulfate, and/or mutational analysis confirming pathogenic variants in the *HGSNAT* gene. HGSNAT enzyme activity was assessed in leukocytes using a fluorometric assay with a specific substrate, as previously reported [15] or in dried blood spot on filter paper, as previously described [13]. Molecular genetics analysis of the *HGSNAT* gene was performed on DNA extracted from peripheral blood using next-generation sequencing on the Ion Torrent S5 platform with a pre-validated NGS panel [14]. Patients’ medical records were reviewed for biochemical findings, medical history, clinical manifestations, and assessments. The study was approved by the Ethics Committee of the Hospital de Clínicas de Porto Alegre (GPPG: 03-066). Written informed consent was obtained from a parent in the case of minors or patients with cognitive impairment, or directly from patients aged 18 years or older, authorizing also the use of images.

### 2.2. Statistical Analysis

Data were entered into a Microsoft Excel spreadsheet. Statistical analyses were performed using SPSS version 12.0 for Windows (SPSS Inc., Chicago, IL, USA). Results are presented as descriptive data. Correlations between two variables were assessed using Pearson’s correlation test. A *p*-value < 0.05 was considered statistically significant.

## 3. Results

By 2025, the total number of confirmed MPS IIIC cases was 101 (Table 1). Among these patients, 60% (*n* = 59) were female and 40% (*n* = 42) were male. All patients were from Brazil. The geographic distribution of patients is shown in Figure 1.

The mean age at diagnosis was 11.7 ± 6.92 years. Regarding the mean diagnostic age over the years, from 1983 to 2005 the average age at diagnosis was 9.25 years (*n* = 12); from 2006 to 2015 it increased to 11.74 years (*n* = 43); and from 2016 to 2025 the mean diagnostic age was 12.97 years (*n* = 34). From the diagnosed patients, only 13.5% (*n* = 13) were aged 5 years or younger at the time of diagnosis (Figure 2A). No correlation was found between age at diagnosis and patient gender (*p* > 0.05, *n* = 98). The mean age at symptom onset was 5.39 ± 3.91 years. The difference between mean age at symptom onset and mean age at diagnosis was statistically significant (*p* < 0.05, *n* = 26), demonstrating an average diagnostic delay of approximately 6 years (Figure 2B). Surprisingly, age of diagnosis of patients who had family members previously diagnosed with MPS IIIC did not show significant differences compared to the overall study population (*p* > 0.05, *n* = 26). Twenty-eight parents of MPS IIIC patients reported consanguineous marriage.

Diagnostic confirmation tests showed reduced HGSNAT enzymatic activity (0.11 ± 0.15 nmol/17 h/mg) (Figure 3A) and elevated urinary GAG concentration (226 ± 125.3 μg/mg creatinine—reference levels are age-dependent) (Figure 3B). No correlations were observed between age at diagnosis and HGSNAT enzyme activity (*p* > 0.05, *n* = 95), between uGAGs and HGSNAT enzyme activity (*p* > 0.05, *n* = 95), or between uGAGs and age at diagnosis (*p* > 0.05, *n* = 79).

In twenty-eight percent (*n* = 28) of patients we had sample available and it was possible to perform molecular genetics investigation, with pathogenic variations being identified. The most frequent variants (*n* = 4 alleles) were c.372-2A>G and c.252dupT (p.Val176fs), followed by c.1169delG (p.Trp390fs) (*n* = 3), c.1348delG (p.Asp450fs) (*n* = 3), c.164T>A (p.Leu55Ter) (*n* = 3), IVS10-2A>C (*n* = 3), c.133_134insA (p.Arg45fs) (*n* = 2), and c.376G>T (p.Glu126Ter) (*n* = 2). Additionally, the following variants were each observed in a single patient: c.1170del (p.Trp390Cysfs*17), c.1225G>C (p.Gly409Arg), c.1301G>A (p.Cys434Tyr), c.1464+1G>A, c.1757 (p.Ser586Phe), c.234+1G>A, c.373-2A>G, and c.710C>A (p.Pro237Gln). The localization and structural alterations of these pathogenic variants are illustrated in Figure 4 and Table 1.

The most frequent initial symptoms associated with MPS IIIC were developmental delay (82%), facial dysmorphism (80%), hepatosplenomegaly (25%), macrocephaly (20%), umbilical hernia (13%), joint contractures (12%), recurrent respiratory problems (9%), hearing impairment (7%), and bone dysplasia (6%) (Figure 5 and Table 2). In most cases, clinical diagnostic hypotheses correctly suggested mucopolysaccharidosis type III (90%), followed by unspecified mucopolysaccharidosis (40%), mucopolysaccharidosis type II (10%), and adrenoleukodystrophy (10%) (Table 3). Main behavioral symptoms reported were agitation (61.5%), aggressive behavior (38.5%), hyperactivity (20%), and autistic behavior (7%) (Table 1). The following medications were most commonly used for management of symptoms: carbamazepine (23%), Chlorpromazine (20%), Risperidone (17%), Periciazine (8.5%), Haloperidol (8.5%), Fluoxetine (6%), Amitriptyline (6%) and Valproic Acid (6%) (Table 4).

Across Brazilian regions, we observed significant differences in the mean age at diagnosis, patient journey, and prevalence of consanguinity (Table 5). The patient journey was defined as the interval between the mean age at symptom onset and the mean age at diagnosis. Among regions with more than one representative patient, the North region had the lowest mean age at diagnosis (6.25 years, *n* = 4) and the shortest patient journey (3.25 years), followed by the South (10.2 years, *n* = 19; patient journey of 4.7 years), the Northeast (12 years, *n* = 31; patient journey of 7.7 years), and the Southeast (13 years; patient journey of 8.8 years). Regarding consanguinity, the highest prevalence was found in the Northeast (54% of reported unions, *n* = 24), followed by the Southeast (43%, *n* = 16) and the South (37.5%, *n* = 8).

## 4. Discussion

We performed a retrospective analysis of patients diagnosed with MPS IIIC through the MPS Brazil Network, including data from 101 individuals. According to previous research by our group, the incidence of MPS IIIC in Brazil is 0.07 per 100,000 live births [20], a rate comparable to that reported in Australia (0.07:100,000) and Taiwan (0.03:100,000) [21,22]. Conversely, higher incidences have been described in France (0.15:100,000) [23], Portugal (0.12:100,000), the Czech Republic (0.42:100,000), and the Netherlands (0.21:100,000) [21]. Among the aforementioned studies, relevant comparisons were identified regarding the populations of patients with MPS IIIC. Considering the number of patients born in Brazil during the time period, we reach a minimal incidence of 0.09 in 100,000.

Besides frequency, other comparisons can be made: one of them refers to the age at diagnosis. In our cohort, the mean age at diagnosis was 11.7 ± 6.92 years. Similar results were reported in earlier studies, such as 12 years in France [23] and in the Netherlands [24]. More recent investigations, however, indicate a substantial reduction in diagnostic age, with reports of 4 years (range: 2–6) in Colombia [25], 7.6 ± 4.5 years in China [26], and similar findings in Korea [27]. Kuiper et al. [28] reported that the diagnostic odyssey remains high (33 months), highlighting the need for standardized guidelines and diagnostic workflows to address this issue. Our findings indicate a marked diagnostic delay in the Brazilian population, with values comparable to those reported in previous decades, despite the fact that diagnostic hypotheses proposed by Brazilian physicians were predominantly related to Mucopolysaccharidosis III (90%). Most of the physicians from the MPS Brazil network are medical geneticists, which suggests that knowledge of MPS III among this group is high, but it usually takes a lot of time to have these patients referred to a geneticist, possibly due to the unspecific symptoms. We searched for explanations for the diagnostic delay, but, surprisingly, we could not find differences in the time to diagnosis when comparing less developed regions of the country, such as the Northeast, with regions that have more resources, such as the Southeast. The time when patients were diagnosed also did not prove to be different, as the results for recently diagnosed patients are similar to those for patients diagnosed three or four decades ago.

In our study, the mean age at symptom onset was 5.39 ± 3.91 years, suggesting that the Brazilian cohort presents a comparatively milder phenotype. This is later than the averages reported in other countries, such as the Netherlands (3.5 years, range: 1–6) [24], Greece (2.5 years) [23], Korea (2.8 ± 0.8 years) [27], China (4.2 ± 2.5 years) [26], and the earlier of 4 years [29]. These data may also explain in part the apparent “diagnostic delay” observed in Brazil or may occur as a consequence of it.

In our cohort, the main findings in later symptom onset patients were primarily characterized by neurodevelopmental delay (82%), facial dysmorphism (80%), and hepatosplenomegaly (25%). By contrast, previous studies describing earlier onset patients reported a predominance of neurodevelopmental delay, behavioral problems, and diarrhea [24]. Behavioral manifestations, emphasized in the literature as highly prevalent [24,26,27], were observed in 25% of our patients. This discrepancy may reflect a comparatively milder phenotype in our population, or the fact that in many cases, we might not have a complete description of the symptoms. The fact that gastrointestinal manifestations, frequently reported in MPS IIIC populations worldwide [30,31], were underrepresented in our cohort, could also suggest that.

Among the pathogenic variants identified in Brazilian patients, several novel variants not previously reported in the literature were observed: c.133_134insA (p.Arg45fs), c.373-2A>G, c.1301G>A (p.Cys343Tyr), c.1169delG (p.Trp390fs), IVS10+2A>C, c.378G>T (p.Glu126Ter), c.1225G>C (p.Gly409Arg), c.1757C>T (p.Ser586Phe), c.376G>T (p.Glu126Ter), IVS13-1G>A, and c.1170del (p.Trp390Cysfs*17). Previously reported mutations identified in Brazilian patients include c.1348delG (p.Asp450fs), c.525dupT (p.Val176fs), c.234+1G>A, and c.164T>A (p.Leu55Ter) [16]. The phenotypic manifestations previously observed in these patients are consistent with those described in our cohort. Genotype–phenotype correlations were observed with Portuguese variants, namely c.372-2A>G and c.525dupT (p.Val176Cysfs*16), in relation to patients from our cohort [16,18]. Similarly, the pathogenic variants c.1464+1G>A (reported in Italian patients) [17] and c.710C>A (p.Pro237Gln) (described in Chinese patients) [19] also showed phenotypic features comparable to those observed in the Brazilian population. Although several studies have demonstrated a genotype-phenotype correlation [32,33,34], our statistical analysis did not reveal any significant associations for the novel variants identified in our cohort. Nevertheless, the presence of these variants in Brazil requires further investigation through ancestry-based analyses.

Although no specific treatment is currently available for MPS IIIC [35], several tools for symptomatic management have been described. Among the main clinical concerns is the control of behavioral manifestations. In our cohort, therapeutic approaches included antihistamines, melatonin, chloral hydrate, and benzodiazepines, consistent with previous reports in the literature [36]. As previously reported by our group, achieving homogeneous outcomes in patients with MPS III is not always feasible [4]. A notable example is the strong adherence to fluoxetine use among patients with MPS IIIA [37], whereas in our cohort, adherence to this medication was limited. The use of D1 dopamine receptor antagonists has recently been proposed for managing Autism Spectrum Disorder–related symptoms in these patients [38]; however, further clinical studies are required to confirm this hypothesis.

## 5. Conclusions

This study reports the largest cohort of Brazilian patients with MPS IIIC to date, providing new insights into the epidemiology, clinical features, and genetic spectrum of the disease in the country. Our findings highlight a later mean age at symptom onset and a significant diagnostic delay compared with more recent international reports, although the diagnostic hypotheses raised by physicians were largely accurate. The identification of several novel pathogenic variants expands the known mutational spectrum of *HGSNAT* and underscores the genetic heterogeneity of MPS IIIC in Brazil. Moreover, the genotype–phenotype correlations observed with Portuguese, Italian, and Chinese variants suggest possible founder effects and emphasize the need for ancestry-based analyses to clarify the distribution of these mutations in the Brazilian population. Despite advances in symptomatic management, the absence of disease-modifying therapies remains a critical challenge. It is important to highlight that the results presented in this study are limited by the lack of longitudinal follow-up regarding each patient’s clinical course. A properly designed natural history study will be required to address this point, as the methodological objective of the present report is restricted to the epidemiological characterization of the population, as previously stated. Taken together, our results reinforce the importance of early recognition, biochemical and genetic diagnosis, and continued investment in therapeutic research to improve outcomes for patients with MPS IIIC.

## Figures and Tables

**Figure 1 diseases-14-00005-f001:**
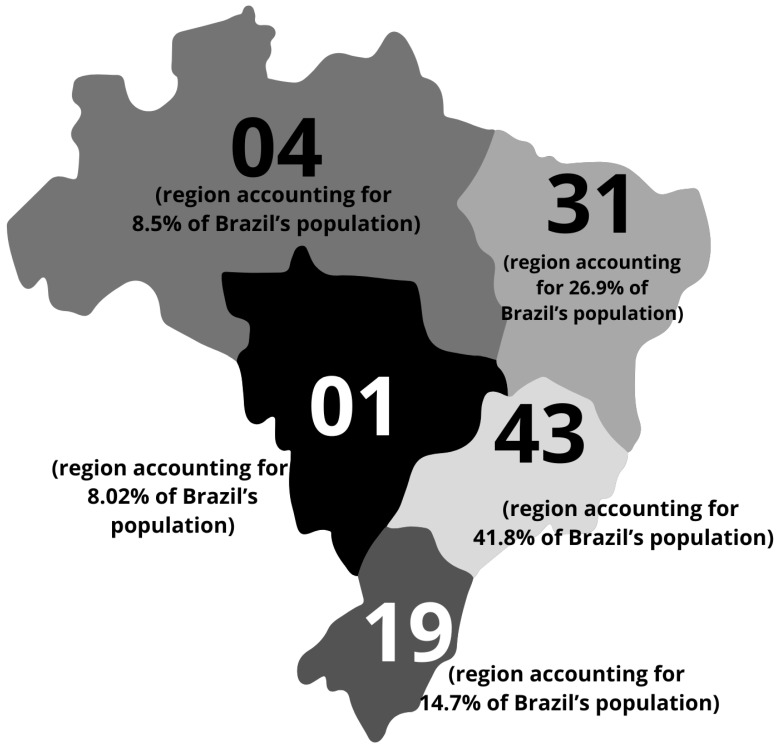
Characteristics of the Brazilian Mucopolysaccharidosis IIIC (MPS IIIC) population. Geographic distribution of patients diagnosed by the MPS Brazil Network across the Brazilian territory. The region with the highest prevalence of cases is the Southwest, accounting for 41.8% of all reported cases, followed by the Northeast (26.9%), South (14.7%), North (8.5%), and Midwest (8.5%).

**Figure 2 diseases-14-00005-f002:**
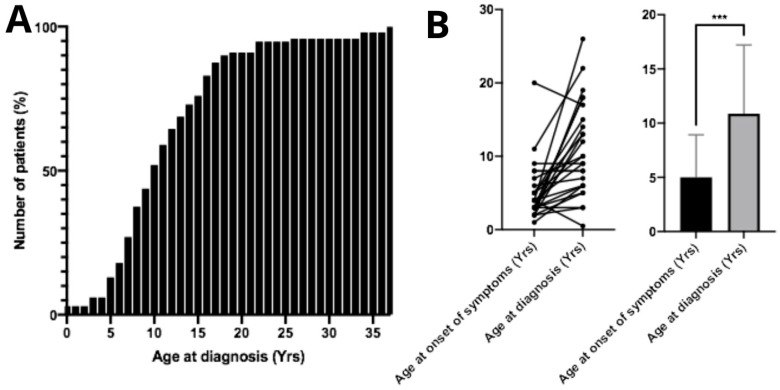
Age at diagnosis and onset of first symptoms in patients with Mucopolysaccharidosis type IIIC. (**A**) Cumulative percentage of patients diagnosed according to age. The mean age at diagnosis among Brazilian patients is 11.7 ± 6.92 years. (**B**) A significant difference (*** *p* < 0.001) was observed between the age of symptom onset and the age at diagnosis, indicating a substantial diagnostic delay in these patients.

**Figure 3 diseases-14-00005-f003:**
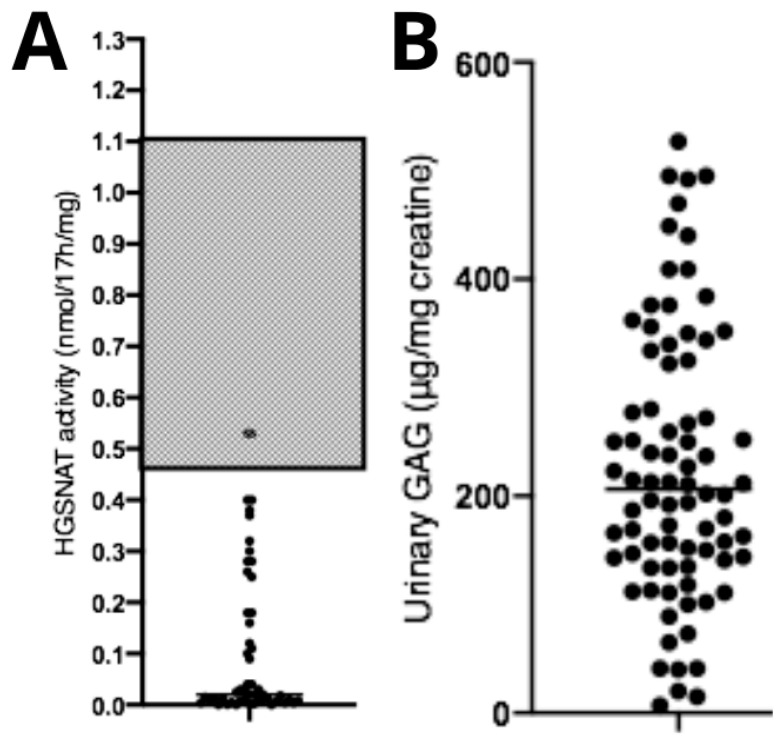
Biochemical data from patients with Mucopolysaccharidosis type IIIC. (**A**) Data showing reduced enzymatic activity compared with the typical reference range (gray area), and (**B**) a significant increase in urinary GAGs compared with normal reference values (not shown in the figure, as they fall below the displayed range and are age-dependent). Dots represent the patients values of uGAGs.

**Figure 4 diseases-14-00005-f004:**
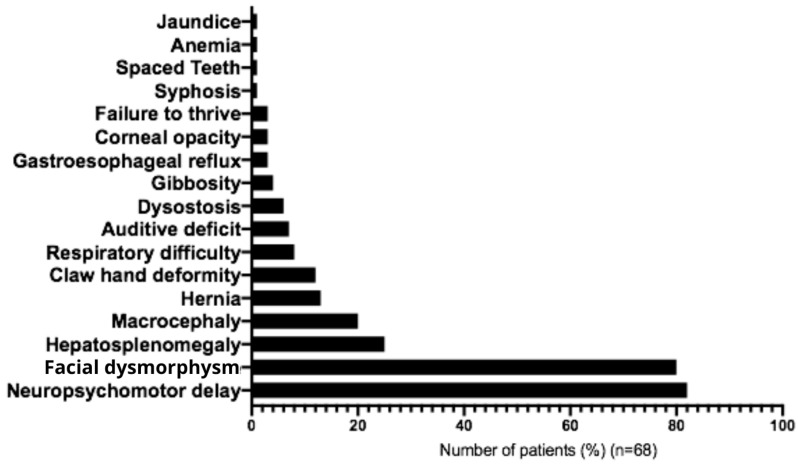
Clinical features observed in Brazilian patients with Mucopolysaccharidosis type IIIC at diagnosis.

**Figure 5 diseases-14-00005-f005:**
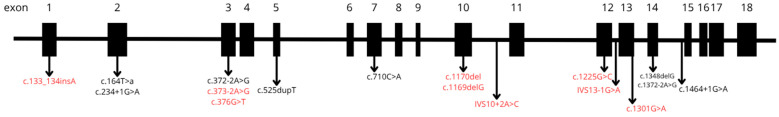
Pathogenic variants identified in patients diagnosed by the MPS Brazil Network. Newly identified variants not previously described in the scientific literature are shown in red.

**Table 1 diseases-14-00005-t001:** Main characteristics of the population of patients with Mucopolysaccharidosis IIIC diagnosed through the MPS Brazil Network.

No	Gender	Current Age (Yrs)	Age at Diagnosis (Yrs)	Birth Place	Creatin (mg/L)	HGSNAT Activity (nmol/17 h/mg) (Reference Value nmol/17 h/mg) *	Variant	Previous Report	Urinary GAG (Age-Related Reference Value) (μg/mg Creatine)	Affected Family Members	Age at Onset of Symptoms (Yrs)	Consanguinity	Age at First Seizure	Reported Behavioral Problems
1	F	37	17	Rio Grande do Sul	-	1.8 (14–81)	-	-	-	-	-	-	-	-
2	M	20	14	São Paulo	2319	0.25 (0.46–1.13)	-	-	384 (13–59)	-	-	-	-	-
3	M	8	4	Alagoas	913	0.01 (0.46–1.13)	c.1348delG (p.Asp450fs)/c.1348delG (p.Asp450fs)	Brazil: Martins et al., 2019 [16]	201 (67–124)	-	-	-	-	-
4	F	28	15	Rio de Janeiro	-	7.3 (14–81)	-	-	144 (26–97)	-	-	1	-	-
5	F	26	17	Pernambuco	947	0.018 (0.46–1.13)	-	-	251 (13–59)	1	-	1	-	-
6	F	-	11	Rio Grande do Sul	508	1.2 (14–81)	-	-	-	-	-	-	-	-
7	F	28	8	Rio Grande do Sul	-	2.3 (14–81)	-	-	-		-	-	-	-
8	M	5	7	Piauí	111	0.40 (0.46–1.13)	**c.133_134insA (p.Arg45fs)/c.133_134insA (p.Arg45fs)**	**Novel variant**	352 (133–274)	0	-	0	-	-
9	M	14	5	Santa Catarina	-	0.009 (0.46–1.13)	c.1464+1G>A/c.164T>A (p.Leu55Ter)	Italy: Fedele et al., 2007 [17]/Brazil: Martins et al., 2019 [16]	211 (53–115)	-	-	-	-	Agitation
10	F	19	10	Ceará	696	0.03 (0.46–1.13)	c.372-2A>G/c.372-2A>G	Portugal: Coutinho et al., 2008 [18]	340 (26–97)	-	-	-	-	-
11	F	26	15	Minas Gerais	654	0.015 (0.46–1.13)	-	-	173 (13–59)	-	5	-	-	-
12	F	37	34	São Paulo	544	7.3 (14–81)	-	-	215 (53–115)	0	-	1	-	-
13	M	18	8	Paraná	1464	0.004 (0.46–1.13)	c.525dupT (p.Val176fs)/c.525dupT (p.Val176fs)	Brazil: Martins et al., 2019 [16]	492 (122–463)	1	8	0	-	-
14	M	-	8	Bahia	835	2.4 (14–81)	-	-	223 (122–463)	0	8	0	-	-
15	M	10	7	Rio Grande do Norte	1455	0.10 (0.46–1.13)	-	-	213 (44–106)	0	-	1	-	Agitation
16	F	33	18	São Paulo	1282	2.3 (14–81)	-	-	-	-	3	0	-	Aggressive behavior
17	F	45	37	Rio de Janeiro	2220	0.18 (0.46–1.13)	-	-	15 (13–45)			-		
18	F	34	22	Paraíba	-	7.4 (14–81)	-	-	-	1	-	0	-	-
19	F	20	17	São Paulo	997	0.12 (0.46–1.13)	**c.133_134insA (p.Arg45fs)/c.133_134insA (p.Arg45fs)**	**Novel variant**	147 (13–59)	-	-	1	-	-
20	M	14	7	São Paulo	387	0.38 (0.46–1.13)	c.710C>A (p.Pro237Gln)/c.234+1G>A	China: Zhao et al., 2024 [19]/Brazil: Martins et al., 2019 [16]	227 (44–106)	-	-	1	-	-
21	M	46	37	Rio de Janeiro	-	0.03 (0.46–1.13)	-	-	-	-	-	-	-	-
22	M	23	11	Rio Grande do Sul	1253	0.03 (0.46–1.13)	-	-	237 (25–97)	-	-	-	-	-
23	M	17	14	Paraíba	625	0.16 (0.46–1.13)	-	-	135 (13–59)	1	3	1	-	-
24	M	23	14	Pernambuco	1140	0.013 (0.46–1.13)	-	-	325 (13–59)	1	-	1	-	-
25	F	20	10	Espírito Santo	854	0.01 (0.46–1.13)	-	-	250 (26–97)	0	5	0	-	Agitation
26	M	44	10	Rio Grande do Sul	586	-	-	-	-	-	-	-	-	-
27	M	25	14	Minas Gerais	739	9.5 (14–81)	-	-	202 (13–53)	-	-	-	-	-
28	M	41	7	Rio Grande do Sul	370	0 (0.03–0.11)	-	-	194 (78–280)	1	6	-	-	Aggressive behavior and Agitation
29	F	17	8	Ceará	667	0.025 (0.46–1.13)	**c.373-2A>G/c.373-2A>G**	**Novel variant**	238 (44–106)	-	-	1	-	-
30	M	13	3	Santa Catarina	617	0.0008 (0.46–1.13)	**c.1301G>A (p.Cys434Tyr)/c.1301G>A (p.Cys434Tyr)**	**Novel variant**	322 (67–124)	0	2	0	-	-
31	F	29	16	Paraíba	857	8.5 (14–81)	-	-	134 (13–57)	0	-	1	-	-
32	F	25	12	São Paulo	-	8.6 (14–81)	-	-	-	-	-	-	-	-
33	F			São Paulo	1236	0.53 (14–81)	-	-	495 (44–106)	-	-	-	-	-
34	F	20	7	Pará	652	6 (14–81)	-	-	157 (44–106)	-	-	1	-	Autistic behavior
35	M	26	13	São Paulo	1757	6.7 (14–81)	-	-	111 (26–97)	-	5	-	-	-
36	M	23	11	São Paulo	1010	0.008 (0.46–1.13)	**c.1169delG (p.Trp390fs)/c.1169delG (p.Trp390fs)**	**Novel variant**	-	-	-	-	-	-
37	M	13	19	Paraná	995	0.28 (0.46–1.13)	-	-	376 (26–97)	-	2	-	12	Agitation and Aggressive behavior
38	M	17	5	Minas Gerais	-	2 (14–81)	c.1348delG (p.Asp450fs)/c.1348delG (p.Asp450fs)	Brazil: Martins et al., 2019 [16]	440 (53–115)	-	3	-	-	Aggressive behavior
39	M	22	8	São Paulo	-	0.09 (0.46–1.13)	**c.1169delG (p.Trp390fs)/c.1169delG (p.Trp390fs)**	**Novel variant**	-	-	-	-	-	Aggressive behavior
40	F	29	16	Paraíba	-	6.1 (14–81)	-	-	134 (13–52)	-	-	-	-	-
41	F	25	12	São Paulo	-	8.6 (14–81)	-	-	267 (78–280)	-	-	-	-	-
42	M	28	13	São Paulo	-	6.7 (14–81)	-	-	111 (26–97)	-	-	-	-	-
43	F	30	8	São Paulo	-	0.53 (14–81)	-	-	495 (44–106)	-	-	-	8	Hyperactivity and Agitation
44	F	21	7	Pará	-	5.7 (14–81)	c.1348delG (p.Asp450fs)/c.1348delG (p.Asp450fs)	Brazil: Martins et al., 2019 [16]	157 (44–106)	-	-	-	-	Autistic behavior
45	M	22	11	São Paulo	-	0.008 (0.46–1.13)	**c.1169delG (p.Trp390fs)/c.1169delG (p.Trp390fs)**	**Novel variant**	41 (10.34)	-	-	-	-	-
46	M	19	13	Mato Grosso do Sul	-	0.28 (0.46–1.13)	-	-	376 (26–97)	-	2y6m	-	12	Agitation and Aggressive behavior
47	M	6	0.5	Pernambuco	-	0.26 (0.46–1.13)	-	-	527 (133–460)	0	4m	0	-	-
48	F	15	16	Minas Gerais	295	10.3 (14–81)	c.372-2A>G/c.372-2A>G	Portugal: Coutinho et al., 2008 [18]	362 (13–55)	-	-	-	-	-
49	M	25	10	São Paulo	-	7.6 (14–81)	-	-	65 (11–37)	0	-	0	-	-
50	M	18	16	Paraíba	611	2.2 (14–81)	**IVS10+2A>C/IVS10+2A>C**	**Novel variant**	150 (26–97)	0	-	0	-	-
51	M	9	10	Paraíba	312	0.014 (0.46–1.13)	-	-	356 (26–97)	-	7	1	-	-
52	F	18	8	Bahia	142	0.18 (0.46–1.13)	c.525dup (p.Val176Cysfs*16)/c.525dup (p.Val176Cysfs*16)	Portugal: Martins et al., 2019 [16]	277 (44–106)	0	-	0	Presence	Agitation
53	M	29	18	São Paulo	754	0.0 (0.46–1.13)	**IVS10+2A>C/IVS10+2A>C**	**Novel variant**	118 (53–59)	1	3	-	-	Agitation
54	M	37	34	Ceará	-	-	-	-	-	-	-	-	-	-
55	F	33	16	Rio de Janeiro	-	2.2 (14–81)	-	-	-	-	-	-	-	-
56	F	33	16	Rio de Janeiro	-	2.0 (14–81)	-	-	-	-	-	-	-	-
57	M	32	22	Rio Grande do Sul	1426	0.004 (0.46–1.13)	**IVS10+2A>C/IVS10+2A>C**	**Novel variant**	100 (13–45)	-	11	1	-	Agitation
58	M	36	18	São Paulo	642	4.4 (14–81)	-	-	212 (26–97)	-	-	-	-	-
59	F	20	6	Rio Grande do Sul	1561	6.4 (14–81)	-	-	252 (53–115)	-	4	-	-	Hyperactivity (4y)
60	F	24	15	Pernambuco	1679	0.010 (0.46–1.13)	-	-	196 (13–59)	-	-	1	-	-
61	F	9	5	Santa Catarina	1844	0.02 (0.46–1.13)	-	-	170 (53–115)	-	-	1	-	Agitation
62	F	32	16	Paraíba	-	5 (14–81)	**c.376G>T (p.Glu126Ter)/c.376G>T (p.Glu126Ter)**	**Novel variant**	-	-	-	-	-	-
63	M	35	10	São Paulo	1031	1.67 (14–81)	c.164T>A (p.Leu55Ter)/c.164T>A (p.Leu55Ter)	Brazil: Martins et al., 2019 [16]	112 (11–37)	-	-	1	-	Aggressive behavior
64	F	9	5	São Paulo	430	0.4 (0.46–1.13)	-	-	187 (53–115)	-	-	1	1y5m	Agitation
65	M	23	12	São Paulo	1067	0.0 (0.46–1.13)	**c.1225G>C (p,Gly409Arg)/c.1757C>T (p.Ser586Phe)**	**Novel variant/Novel variant**	169 (26–97)	-	-	0	Presence	-
66	M	21	7	São Paulo	1617	7.2 (14–81)	-	-	102 (44–106)	-	-	1	-	-
67	M	9	5	Paraíba	-	0.01 (0.46–1.13)	-	-	-	1	-	-	-	-
68	M	10	6	Pernambuco	625	0.002 (0.46–1.13)	-	-	259 (53–115)	-	1	-	4	Hyperactivity
69	M	26	-	-	-	-	**c.376G>T (p.Glu126Ter)/c.376G>T (p.Glu126Ter)**	**Novel variant**	-	-	-	-	-	-
70	M	33	11	Paraíba	-	1.5 (14–81)	-	-	240 (44–106)	-	-	1	-	-
71	M	40	9	São Paulo	269	1.9 (19–85)	-	-		-	9	0	-	-
72	M	19	6	Minas Gerais	226	18 (31–110)	-	-	470 (53–115)	-	-	-	-	-
73	F	11	8	Pará	757	-	-	-	-	1	-	0	-	
74	F	14	9	Santa Catarina	1092	0.32 (0.46–1.13)	-	-	89 (26–97)	0	-	0	-	-
75	F	21	8	Paraíba	288	0.02 (0.46–1.13)	-	-	272 (44–106)	-	-	0	-	-
76	F	18	3	Pará	1184	4.8 (5.5–24)	**IVS13-1G>A/IVS13-1G>A**	**Novel variant**	166 (67–124)	1	3	1	Presence	Agitation and Hyperactivity
77	F	25	9	Alagoas	-	3.7 (14–81)	-	-	158 (26–97)	0	-	0	-	-
78	F	34	5	São Paulo	674	4 (14–81)	-	-	40 (3.4–11)	1	2	0	-	Agitation
79	M	17	8	Paraíba	-	5.7 (14–81)	-	-	180 (44–106)	-	-	-	-	-
80	M	13	11	Bahia	230	0.11 (0.46–1.13)	**c.1170del (p.Trp390Cysfs*17)/c.1170del (p.Trp390Cysfs*17)**	**Novel variant**	409 (44–106)	1	-	1	-	Agitation
81	M	25	12	Paraíba	2311	6.2 (14–81)	-	-	73 (26–52)	-	3	-	-	-
82	M	20	7	São Paulo	767	0.005 (0.46–1.13)	-	-	141 (44–106)	-	-	-	-	-
83	M	15	10	Paraná	759	0.30 (0.46–1.13)	-	-	41 (26–97)	1	-	0	-	-
84	M	47	11	Rio Grande do Sul	1089	7.7 (14–81)	-	-	7/9 (3.4–11)	-	-	1	-	-
85	M	14	6	São Paulo	-	0.004 (0.46–1.13)	c.164T>A (p.Leu55Ter)/c.372-2A>G	Brazil: Martins et al., 2019 [16]/Coutinho et al., 2008 [18]	-	-	3	-	3	Agitation and Aggressive behavior
86	F	29	12	Rio de Janeiro	1121	3.8 (14–81)	-	-	280 (26–97)	-	-	-	-	-
87	M	30	20	Pernambuco	272	0.013 (0.46–1.13)	-	-	334 (13–46)	0	-	1	-	-
88	F	5	4 m	Rio Grande do Sul	332	34 (58–242)	-	-	449 (133–460)	-	-	-	-	-
89	M	12	5	São Paulo	476	0.04 (0.46–1.13)	-	-	213 (44–106)	-	-	-	-	-
90	M	14	1 m	Espírito Santo	146	5.3 (14–81)	-	-	-	-	-	-	1 m	-
91	M	35	26	São Paulo	849	0.007 (0.46–1.13)	-	-	163 (13–45)	0	4	0	1 y	Agitation and Aggressive behavior
92	M	29	10	Ceará	-	2.7 (14–81)	c.525dupT (p.Val176fs)/c.525dupT (p.Val176fs)	Portugal: Martins et al., 2019 [16]	250 (26–97)	-	-	0	-	-
93	F	10	5	Rio Grande do Sul	1610	0.02 (0.46–1.13)	-	-	152 (53–115)	-	-	-	-	-
94	M			-		8 (14–81)	-	-	-	-	-	-	-	-
95	M	30	9	Paraíba	336	1.8 (14–81)	-	-	350 (26–97)	1	-	1	-	-
96	M	27	13	São Paulo	1626	8.9 (14–81)	-	-	113 (26–97)	-	-	-	-	-
97	F	13	9	Bahia	344	0.04 (0.46–1.13)	-	-	143 (44–106)	0	4	1	-	-
98	M	30	6	São Paulo	348	2.7 (14–81)	-	-	20 (5.7–13)	0	-	0	-	Hyperactivity and Aggressive behavior
99	M	26	17	Paraná	808	0.002 (0.46–1.13)	c.525dupT (p.Val176fs)/c.525dupT (p.Val176fs)	Brazil: Martins et al., 2019 [16]	344 (13–59)	1	20	0	-	-
100	F	6	3	Espírito Santo	261	0.37 (0.46–1.13)	-	-	192 (67–124)	-	-	-	-	-
101	M	21	9	São Paulo	-	3.7 (14–81)	-	-	409 (40–370)	-	-	-	-	-

* Reference values change according to the source of material. Bold indicated novel variants.

**Table 2 diseases-14-00005-t002:** Signs and Symptoms in Brazilian patients with Mucopolysaccharidosis IIIC.

Symptoms	Number of Patients (%)
Neurodevelopmental delay	56/58 (82%)
Coarse facies	54/68 (80%)
Hepatosplenomegaly	17/68 (25%)
Macrocephaly	14/68 (20%)
Hernia	09/68 (13%)
Claw hand deformity	08/68 (12%)
Respiratory difficulty	06/68 (9%)
Auditive deficit	05/68 (7%)
Dysostosis	04/68 (6%)
Gibbus	03/68 (4%)
Gastroesophageal reflux	02/68 (3%)
Corneal opacity	02/68 (3%)
Failure to thrive	02/68 (3%)
Kyphosis	01/68 (1%)
Spaced teeth	01/68 (1%)
Anemia	01/68 (1%)
Jaundice	01/68 (1%)

**Table 3 diseases-14-00005-t003:** Initial diagnostic hypotheses on Brazilian patients with Mucopolysaccharidosis type IIIC.

Age at Diagnosis	
Mean	11 y
Range	1 y–37 y
**Consanguineous parents (** ** *n* ** **)**	
Proportion	25/49
**Diagnostic hypotheses**	
Mucopolysaccharidosis type III	20/22 (90%)
Mucopolysaccharidoses not specified	09/22 (40%)
Mucopolysaccharidosis type II	02/22 (10%)
Adrenoleukodystrophy	02/22 (10%)
Mucopolysaccharidosis type IV	01/22 (5%)
Glycoproteinoses	01/22 (5%)
Niemann Pick type C	01/22 (5%)
Usher syndrome	01/22 (5%)
Mucopolysaccharidosis type I	01/22 (5%)
Mucopolysaccharidosis type VI	01/22 (5%)

**Table 4 diseases-14-00005-t004:** Main medications prescribed for symptomatic management in patients with MPS IIIC.

Drug	Number of Patients (%)
Carbamazepine	08/35 (23%)
Chlorpromazine	07/35 (20%)
Risperidone	06/35 (17%)
Periciazine	03/35 (8.5%)
Haloperidol	03/35 (8.5%)
Fluoxetine	02/35 (6%)
Amitriptyline	02/35 (3%)
Valproic Acid	02/35 (6%)
Vancomycin	01/35 (3%)
Omeprazole	01/35 (3%)
Vitamin B Complex	01/35 (3%)
Zinc supplement	01/35 (3%)
Levomepromazine	01/35 (3%)
Biperiden	01/35 (3%)
Periciazine	01/35 (3%)
Clonazepam	01/35 (3%)

**Table 5 diseases-14-00005-t005:** Regional differences in Brazil regarding to diagnosis of MPS IIIC.

Region	Mean Age at Diagnosis (Yrs) (Number of Patients)	Mean Age at Onset of Symptoms (Yrs) (Number of Patients)	Patient Journey *	Prevalence of Consanguinity (%) (Number of Patients)
Midwest	13 (1)	2 (1)	11	0% (0)
North	6.25 (4)	3 (1)	3.25	100% (2)
Northeast	12 (31)	4.3 (7)	7.7	54% (24)
South	10.2 (19)	5.5 (6)	4.7	37.5% (8)
Southeast	13 (45)	4.2 (10)	8.8	43% (16)

* Patient Journey = (Media age at diagnosis) − (media age at onset of symptoms).

## Data Availability

The data presented in this study are available on request from the corresponding author.

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
