# Peer review of "Unveiling Mucopolysaccharidosis IIIC in Brazil: Diagnostic Journey and Clinical Features of Brazilian Patients Identified Through the MPS Brazil Network"

_diseases, 2025, doi:10.3390/diseases14010005_

Round 1

Reviewer 1 Report

Comments and Suggestions for Authors

 This manuscript is titled:

“Unveiling Mucopolysaccharidosis IIIC in Brasil: Diagnostic 2 Journey and Clinical Features of Brazilian Patients Identified 3 Through the MPS Brazil Network”. This is the largest natural history study done, to date, in Brazil for MPS IIIC.'

This manuscript describes a careful study of retrospective natural history data from the MPS Brazil Network on the very rare condition called mucopolysaccharidosis type IIIC (Sanfilippo syndrome type C, or MPS IIIC), which is cause by inherited pathogenic mutations to the HGSNAT gene.   Clinical, biochemical and genetic data were analyzed from 101 patients.  This is the largest natural history study done, to date, in Brazil for MPS IIIC.

Important findings include recurrent alleles in the genetic data were (c.372-2A>G, c.252dupT), identification of several novel variants, genotype-phenotype similarities when comparing Brazilian, Chinese and Italian populations, and a diagnostic delay of approximately 6 years.  The authors found that there was a diagnostic delay of around 6 years, in many cases, even if other family members had previously been diagnosed with MPS IIIC.  However, the authors were able to identify some regional differences in the “patient journey” (time from first noted symptom to diagnosis). The authors were able to capture data on the most common initial presenting symptoms, common behavioral abnormalities, the most commonly used medications in this population.

Incidence of MPS IIIC in Brazil was found to be comparable to Taiwan and Australia, but less than the incidence in France, Netherlands and Czech Republic.  The authors were also able to compare reported symptom onset age for the Brazilian population to other countries and found that symptoms onset was report a few years earlier in some countries (Netherlands, Greece, Korea, China).  The reason for differences in reported symptom onset are not completely understood, but the authors propose this may reflect a possibly slightly milder phenotypic presentation in the Brazilian population possible related to the most prevalent 2 mutations found in patients in Brazil. 

In summary, this research is a very important contribution to the natural history data for the very rare MPS IIIC disease.  The researchers have captured important observations regarding phenotype, genetic data, delays in diagnosis, and have also provided very helpful comparisons to data from numerous other countries in the world. Such natural history data is critical for effective therapy development and improving understanding of approaches to disease diagnosis and management in a global community in the unmet need for effective treatment is an ongoing and time-sensitive concern. In addition, the data provided through this project will contribute significantly to future assessments of newborn screening concerns for MPS CIII.

Author Response

Dear Editor

We sincerely thank the reviewers for their valuable and insightful contributions to this work. Below, we provide our point-by-point responses to each reviewer, highlighted in yellow.

REVIEWER 1

This manuscript is titled: “Unveiling Mucopolysaccharidosis IIIC in Brasil: Diagnostic 2 Journey and Clinical Features of Brazilian Patients Identified 3 Through the MPS Brazil Network”. This is the largest natural history study done, to date, in Brazil for MPS IIIC.'

This manuscript describes a careful study of retrospective natural history data from the MPS Brazil Network on the very rare condition called mucopolysaccharidosis type IIIC (Sanfilippo syndrome type C, or MPS IIIC), which is cause by inherited pathogenic mutations to the HGSNAT gene.   Clinical, biochemical and genetic data were analyzed from 101 patients.  This is the largest natural history study done, to date, in Brazil for MPS IIIC.

Important findings include recurrent alleles in the genetic data were (c.372-2A>G, c.252dupT), identification of several novel variants, genotype-phenotype similarities when comparing Brazilian, Chinese and Italian populations, and a diagnostic delay of approximately 6 years.  The authors found that there was a diagnostic delay of around 6 years, in many cases, even if other family members had previously been diagnosed with MPS IIIC.  However, the authors were able to identify some regional differences in the “patient journey” (time from first noted symptom to diagnosis). The authors were able to capture data on the most common initial presenting symptoms, common behavioral abnormalities, the most commonly used medications in this population.

Incidence of MPS IIIC in Brazil was found to be comparable to Taiwan and Australia, but less than the incidence in France, Netherlands and Czech Republic.  The authors were also able to compare reported symptom onset age for the Brazilian population to other countries and found that symptoms onset was report a few years earlier in some countries (Netherlands, Greece, Korea, China).  The reason for differences in reported symptom onset are not completely understood, but the authors propose this may reflect a possibly slightly milder phenotypic presentation in the Brazilian population possible related to the most prevalent 2 mutations found in patients in Brazil.

In summary, this research is a very important contribution to the natural history data for the very rare MPS IIIC disease.  The researchers have captured important observations regarding phenotype, genetic data, delays in diagnosis, and have also provided very helpful comparisons to data from numerous other countries in the world. Such natural history data is critical for effective therapy development and improving understanding of approaches to disease diagnosis and management in a global community in the unmet need for effective treatment is an ongoing and time-sensitive concern. In addition, the data provided through this project will contribute significantly to future assessments of newborn screening concerns for MPS CIII.

RESPONSE: We sincerely appreciate the reviewers’ comments on our manuscript.

Reviewer 2 Report

Comments and Suggestions for Authors

This is a very well-written manuscript that summarizes a retrospective study of MPS IIIC patients diagnosed by the MPS Brazil Network between 1983 and 2025. The authors evaluate numerous biochemical and clinical evidence. Several new findings resulted from this study, including: 1) identification of 101 MPS IIIC patients; 2) the average age of symptom onset; 3) the average age of diagnosis; 4) biochemical endpoints (enzyme, GAGs); 5) mutation identification for a subset of patients, some of which represent novel mutations as well as recurring mutations; 6) clinical symptoms; 7) initial diagnoses; 8) symptomatic management; and 9) regional differences in diagnosis, age of onset, and consanguinity. Key novel findings include a substantial gap between the age of onset and the age of diagnosis, the possibility of  milder phenotypes in the Brazilian cohort compared to other MPS IIIC cohorts globally, and the identification of several new HGSNAT gene variants. The results are well-explained and represented well in the associated figures and tables. My only criticism is that the text in Figures 1 & 2 is quite small and would be improved if the font size was increased. 

Author Response

REVIEWER 2

This is a very well-written manuscript that summarizes a retrospective study of MPS IIIC patients diagnosed by the MPS Brazil Network between 1983 and 2025. The authors evaluate numerous biochemical and clinical evidence. Several new findings resulted from this study, including: 1) identification of 101 MPS IIIC patients; 2) the average age of symptom onset; 3) the average age of diagnosis; 4) biochemical endpoints (enzyme, GAGs); 5) mutation identification for a subset of patients, some of which represent novel mutations as well as recurring mutations; 6) clinical symptoms; 7) initial diagnoses; 8) symptomatic management; and 9) regional differences in diagnosis, age of onset, and consanguinity. Key novel findings include a substantial gap between the age of onset and the age of diagnosis, the possibility of  milder phenotypes in the Brazilian cohort compared to other MPS IIIC cohorts globally, and the identification of several new HGSNAT gene variants. The results are well-explained and represented well in the associated figures and tables. My only criticism is that the text in Figures 1 & 2 is quite small and would be improved if the font size was increased.

RESPONSE: We sincerely appreciate the comments provided regarding our work. As suggested, we made significant improvements to the figures, opting to divide them in order to enhance readability, as recommended.

Reviewer 3 Report

Comments and Suggestions for Authors

This study conducted a retrospective analysis of 101 Brazilian patients with MPS IIIC through the MPS Brazil Network, providing one of the largest global cohort datasets and filling a research gap in Latin America for this rare disease. It holds significant clinical and scientific value. While the overall study design is sound and the data comprehensive, certain details could be optimized. Specific recommendations are as follows:

  1. Molecular genetic testing covers only 28% of patients, a relatively low proportion that limits the comprehensive interpretation of genetic characteristics among Brazilian MPS IIIC patients. It is recommended to supplement genetic testing data from additional patients.
  2. The analysis of reasons for diagnostic delays lacks depth, merely speculating that they are related to nonspecific symptoms and the time required for referral to geneticists. It fails to specifically explore factors such as clinicians' knowledge levels and regional disparities in healthcare resources. Additional relevant research or analysis could be conducted.
  3. Some clinical data integrity issues exist, such as missing information for certain patients regarding current age, place of birth, age at symptom onset, and other details. Additionally, data gaps are present for some laboratory indicators (e.g., creatinine levels), which may compromise the comprehensiveness of the results analysis.
  4. The article contains numerous tables; it is recommended to enhance the explanatory text accompanying figures and tables.

     5. Although the article provides a wealth of data, certain information—such as regional        variations and symptom distribution—would be more intuitive and accessible if presented in graphical formats.It is recommended to incorporate visual aids such as bar charts, pie charts, or maps.

Author Response

REVIEWER 3

We sincerely appreciate the reviewer’s comments and suggestions regarding our work.

This study conducted a retrospective analysis of 101 Brazilian patients with MPS IIIC through the MPS Brazil Network, providing one of the largest global cohort datasets and filling a research gap in Latin America for this rare disease. It holds significant clinical and scientific value. While the overall study design is sound and the data comprehensive, certain details could be optimized. Specific recommendations are as follows:

Molecular genetic testing covers only 28% of patients, a relatively low proportion that limits the comprehensive interpretation of genetic characteristics among Brazilian MPS IIIC patients. It is recommended to supplement genetic testing data from additional patients.

RESPONSE: With respect to the molecular testing of Brazilian patients, we agree it would be very important to perform molecular analysis of all our patients. However, it is worth noting that our records date back to the 1990s, and in many cases we lost track of such patients along the years, making it impossible to collect sample for DNA analysis. We performed DNA analysis of all patients whose samples were available.  

The analysis of reasons for diagnostic delays lacks depth, merely speculating that they are related to nonspecific symptoms and the time required for referral to geneticists. It fails to specifically explore factors such as clinicians' knowledge levels and regional disparities in healthcare resources. Additional relevant research or analysis could be conducted.

RESPONSE: We appreciate the methodological suggestion. We tried to analyze some reasons for the discrepancy, but the data we have available is insufficient to do it with robustness. Some of the reasons for that are, for example, the fact that some less-developed areas, such as the north of the country, have only 4 patients, and they were all diagnosed relatively early, even earlier than in more developed areas, such as southeast. Yet, we agree with the reviewer and made some modifications to the text, suggesting some reasons for our findings.

The new version now reads

“We searched for explanations for the diagnostic delay, but, surprisingly, we couldn't find differences in the time to diagnosis when comparing less developed regions of the country, such as the Northeast, with regions that have more resources, such as the Southeast. The time when patients were diagnosed also didn't prove to be different, as the results for recently diagnosed patients are similar to those for patients diagnosed three or four decades ago”

Some clinical data integrity issues exist, such as missing information for certain patients regarding current age, place of birth, age at symptom onset, and other details. Additionally, data gaps are present for some laboratory indicators (e.g., creatinine levels), which may compromise the comprehensiveness of the results analysis.

RESPONSE: We also thank the reviewer for the additional comments. We would like to highlight that our dataset spans a 40-year period. Many of the biochemical analyses available were those feasible at the time each diagnosis was performed, thus reflecting the limitations of their respective eras. In addition, our records do not include longitudinal follow-up, as patients must return to diagnostic centers for reassessment, a process that falls outside the scope and objectives of the MPS Brazil Network. Given the country’s continental dimensions, certain data are inherently difficult to monitor. For these reasons, we explicitly stated the limitations of our study in the conclusion section:

“It is important to highlight that the results presented in this study are limited by the lack of longitudinal follow-up regarding each patient’s clinical course. A future natural history study will be required to address this point, as the methodological objective of the present study is restricted to the epidemiological characterization of the population, as stated.”

The article contains numerous tables; it is recommended to enhance the explanatory text accompanying figures and tables.

RESPONSE: As suggested by the reviewer, we have added more descriptive and detailed captions to our figures and tables.

  1. Although the article provides a wealth of data, certain information—such as regional variations and symptom distribution—would be more intuitive and accessible if presented in graphical formats. It is recommended to incorporate visual aids such as bar charts, pie charts, or maps.

RESPONSE: Despite not following the reviewer’s suggestion to add more graphs, we did highlighted  in the new version of the text some of the topics he mentioned, including a discussion on regional variations. Hopefully the new version is more intuitive.

Reviewer 4 Report

Comments and Suggestions for Authors

Based on the observation that 82% of MPS IIIC patients in the cohort presented with developmental delay, can these patients give informed consent?

Are there any correlations between severity of symptoms and other parameters (age at onset, age at diagnosis, HGSNAT activity, uGAG levels)?

Can you refer the reader to a publication correlating the gene variants with residual HGSNAT activity or disease severity?

Should IVS10+2A>C for patient 50 also be designated as a novel variant, same as for patients 53 and 57?

The reference value for HGSNAT activity is given as 14-81 and 0.46-1.13 nmol/17h/mg. It is unclear what the different values correlated with. It does not appear to be sex or age. Could this be based on the source of material (DBS versus leukocytes)? This needs to be clearly stated.

Author Response

Reviewer 4

Based on the observation that 82% of MPS IIIC patients in the cohort presented with developmental delay, can these patients give informed consent?

RESPONSE: The consent forms for underage patients or those with cognitive impairment were signed by their legal guardians.We included this information in the new version of the manuscript.

Are there any correlations between severity of symptoms and other parameters (age at onset, age at diagnosis, HGSNAT activity, uGAG levels)?

RESPONSE: We ran the correlation tests and observed no correlations between the parameters. This was indicated in the text.

Can you refer the reader to a publication correlating the gene variants with residual HGSNAT activity or disease severity?

RESPONSE: Beesley CE, Jackson M, Young EP, Vellodi A, Winchester BG, 2005, Molecular defects in Sanfilippo syndrome type B (mucopolysaccharidosis IIIB), J Inherit Metab Dis., DOI: 10.1007/s10545-005-0093-y, PMID: 16151907.

Zhao HG, Aronovich EL, Whitley CB, 1998, Genotype-phenotype correspondence in Sanfilippo syndrome type B, Am J Hum Genet., DOI: 10.1086/301682, PMID: 9443875, PMCID: Full Text.

Yogalingam G, Weber B, Meehan J, et al., 2000, Mucopolysaccharidosis type IIIB: characterisation and expression of wild-type and mutant recombinant alpha-N-acetylglucosaminidase and relationship with sanfilippo phenotype in an attenuated patient, Biochim Biophys Acta., DOI: 10.1016/S0925-4439(00)00066-1, PMID: 11068184.

Should IVS10+2A>C for patient 50 also be designated as a novel variant, same as for patients 53 and 57?

RESPONSE: We have made the corrections to the table as recommended. We appreciate the reviewer’s observation.

The reference value for HGSNAT activity is given as 14-81 and 0.46-1.13 nmol/17h/mg. It is unclear what the different values correlated with. It does not appear to be sex or age. Could this be based on the source of material (DBS versus leukocytes)? This needs to be clearly stated.

RESPONSE: The reviewer is right: there are indeed differences when considering the reference values across different testing methods. For this reason, the corresponding reference values are provided in parentheses alongside the results. We also added in the new version a brief explanation for that in the table. Now it reads “* reference values change according to the source of material.

Reviewer 5 Report

Comments and Suggestions for Authors

I kindly request that the term ‘coarse face’ (i.e. figure 1 or table 2) be replaced with a more precise and non-stigmatizing formulation, such as 'facial dysmorphysm' or an objective description of the specific characteristics observed (e.g., thickened facial soft tissues, depressed nasal bridge, broad nasal tip, full lips, or macroglossia) if they have this data.

There is some confusion regarding the total number of patients. In the text and Table 1, a total of 101 patients is reported; however, when grouped by mean age at diagnosis (lines 122 and following), the sum is only 89, even though this field is available for 100 patients. In the same section, Figure 1 shows 43 cases from the ‘southeast’ region, whereas in Table 5, the number of patients is 45.

Since the article reports up to 10 previously undescribed variants, I believe the manuscript would be enriched by providing a more detailed description of the clinical features of those patients, with a specific discussion.

Author Response

Dear Editor

We sincerely thank the reviewers for their valuable and insightful contributions to this work. Below, we provide our point-by-point responses to each reviewer, highlighted in yellow.

REVIEWER 4

I kindly request that the term ‘coarse face’ (i.e. figure 1 or table 2) be replaced with a more precise and non-stigmatizing formulation, such as 'facial dysmorphysm' or an objective description of the specific characteristics observed (e.g., thickened facial soft tissues, depressed nasal bridge, broad nasal tip, full lips, or macroglossia) if they have this data.

RESPONSE: We thank the reviewer for the suggestions. We agree and changed the term accordingly.

There is some confusion regarding the total number of patients. In the text and Table 1, a total of 101 patients is reported; however, when grouped by mean age at diagnosis (lines 122 and following), the sum is only 89, even though this field is available for 100 patients. In the same section, Figure 1 shows 43 cases from the ‘southeast’ region, whereas in Table 5, the number of patients is 45.

RESPONSE: The sample size varies according to the availability of clinical information examined in each section. We strictly followed the scientific rigor of the data available in our records. In some cases, specific values were not reported; therefore, the n for each group may vary depending on the variable being analyzed. This variation reflects the continental dimensions of our country, the heterogeneity in data availability across a 40-year time span, and the differences in information provided by genetic services across Brazil.

Since the article reports up to 10 previously undescribed variants, I believe the manuscript would be enriched by providing a more detailed description of the clinical features of those patients, with a specific discussion.

RESPONSE: Thank you for this particular suggestion. We agree it would be interesting, but unfortunately, we did not have more detailed information available regarding these patients, even after trying to contact some of them or the physicians.

Round 2

Reviewer 5 Report

Comments and Suggestions for Authors

Thank you very much for your response. As a final remark, after reviewing the variants classified as novel, I would like to clarify whether, in case 29 of Table 1, the variant is correctly annotated and whether it corresponds to position c.373 or c.372 to the gene/transcript NM_152419.3, because numbering errors (c.372 vs c.373) can lead to confusion. If the variant is c.373, then it would be appropriate to classify it as novel; however, if it is c.372, it has already been reported in multiple studies https://www.ncbi.nlm.nih.gov/clinvar/variation/1236/

In case 65 there is an error in the description of case 65: “c.1757” does not indicate the nucleotide change, which results in an incomplete annotation. It should be “c.1757C>T”. Same error in line 146.

Please confirm whether the nomenclature c.1170del (p.Trp390Cysfs*17) in case 80 is correct, and whether it is not an equivalent variant misaligned with c.1169delG, which appears in other similar cases.

Author Response

Thank you very much for your response. As a final remark, after reviewing the variants classified as novel, I would like to clarify whether, in case 29 of Table 1, the variant is correctly annotated and whether it corresponds to position c.373 or c.372 to the gene/transcript NM_152419.3, because numbering errors (c.372 vs c.373) can lead to confusion. If the variant is c.373, then it would be appropriate to classify it as novel; however, if it is c.372, it has already been reported in multiple studies https://www.ncbi.nlm.nih.gov/clinvar/variation/1236/

RESPONSE: We appreciate the reviewer's attention to our data. We have reviewed the reports issued by our laboratory and all of them are correctly formatted.

In case 65 there is an error in the description of case 65: “c.1757” does not indicate the nucleotide change, which results in an incomplete annotation. It should be “c.1757C>T”. Same error in line 146.

RESPONSE: Thank you for your feedback. We have corrected the notation in our table.

Please confirm whether the nomenclature c.1170del (p.Trp390Cysfs*17) in case 80 is correct, and whether it is not an equivalent variant misaligned with c.1169delG, which appears in other similar cases.

RESPONSE: The data has been reviewed and our feedback is reflected throughout our text.